# Ovarian Pregnancy: 2 Case Reports and a Systematic Review

**DOI:** 10.3390/jcm12031138

**Published:** 2023-02-01

**Authors:** Zukaa Almahloul, Bedayah Amro, Zuhdi Nagshabandi, Iman Alkiumi, Zeinabs Hakim, Arnaud Wattiez, Muna Tahlak, Philippe R. Koninckx

**Affiliations:** 1Latifa Hospital, Dubai P.O. Box 9115, United Arab Emirates; 2Department of Obstetrics and Gynaecology, University of Strasbourg, 67081 Strasbourg, France; 3Department of OBGYN, Faculty of Medicine, Katholieke University Leuven, 3000 Leuven, Belgium; 4Department of OBGYN, Faculty of Medicine, University of Oxford, Oxford OX1 2JD, UK; 5Department of OBGYN, Faculty of Medicine, University of Cattolica, del Sacro Cuore, 00168 Rome, Italy; 6Department of OBGYN, Faculty of Medicine, Moscow State University, 119991 Moscow, Russia

**Keywords:** ectopic pregnancy, heterotopic pregnancy, laparoscopy, methotrexate, corpus luteum

## Abstract

Ovarian pregnancy is a rare but well-known pathology. However, pathophysiology, diagnosis and treatment are not established. Therefore, all case reports on ovarian pregnancy published in PubMed from November 2011 till November 2022 were reviewed and two case reports were added. In these 84 case reports, 8% of ovarian pregnancies occurred in women without or with blocked oviducts and 23% were localised on the other side than the corpus luteum. Since symptoms are not specific, ovarian pregnancy has to be suspected in all women with abdominal bleeding. Surgical excision is the preferred treatment. However, since an associated intra-uterine pregnancy cannot be excluded, care should be taken not to interrupt this intra-uterine pregnancy with the uterine cannula or by damaging the corpus luteum. In conclusion, in women with abdominal bleeding, an ovarian pregnancy cannot be excluded, even in women with a negative pregnancy test or an empty uterus on transvaginal ultrasonography. Therefore, a laparoscopy is indicated but the surgeon should realise that an associated intra-uterine pregnancy also cannot be excluded and that therefore care should be taken not to interrupt this intra-uterine pregnancy by the uterine cannula or by damaging the corpus luteum.

## 1. Introduction

Ovarian pregnancy could be considered a rare but well-known pathology, considering the 667 articles found in PubMed. Since 1945, the yearly number of articles has been constant at around 10, with 111 articles over the last 10 years, including 12 small series with some narrative reviews [1,2,3,4,5,6].

Ovarian pregnancies are a rare form of ectopic pregnancy, occurring in 0.5% to 1% of ectopic gestations, or one in 7000 to 40,000 live births [1,7]. Although a potentially life-threatening event, the pathophysiology is not understood and the diagnosis remains challenging [8]. The accuracy of the diagnosis with clinical exam, HCG concentrations and ultrasound or MRI imaging is limited [9], with the risk of missing the diagnosis and of overdiagnosing because of cognitive bias [10]. Even the laparoscopic diagnosis can be difficult and many ectopic pregnancies are only diagnosed by pathology after surgery. Therapy is variable and ovarian-sparing surgery is poorly discussed. In addition, the follow-up after treatment is not clear.

Therefore, we decided to perform a systematic review of ovarian pregnancies, triggered by two recent cases of ovarian pregnancy.

## 2. Objectives

This review aimed to understand the pathophysiology of ovarian pregnancies and to improve diagnosis and therapy.

## 3. Materials and Methods

### 3.1. Case Reports

Written informed consent for publication was obtained from both patients. The local institutional review board (IRB) was not required for anonymous case reports.

### 3.2. Systematic Review

#### 3.2.1. Study Selection

PubMed was searched for (“ovarian pregnancy”) OR (“ovarian ectopic pregnancy”) on 5 November 2022. The 111 articles retrieved were screened, 99 were reviewed and 82 case reports were retained. We did not include 5 reviews of imaging, or pathology, and 12 reports in which the information of individual cases could not be retrieved (5 discussed ectopic pregnancies, one did not contain data and 3 were reviews of small series, not permitting extraction of details) (Figure 1).

#### 3.2.2. Data Extraction

These case reports were searched for age and antecedents of the woman such as gravity, parity, abortions, ectopic pregnancies, in-vitro fertilisation (IVF), intra-uterine insemination (IUI), tubal or endometriosis or pelvic surgery, pelvic inflammatory disease (PID) and the use of an intra-uterine device (IUD). For the ovarian pregnancies, we retrieved the duration of the pregnancy, presenting symptoms (pain, vaginal bleeding, shock), HCG concentration, ultrasound and MRI exams, treatment and outcome.

#### 3.2.3. Assessment of Risk of Bias

Since only case reports were reviewed, PROSPERO registration was not performed and the risk of bias was not assessed.

#### 3.2.4. Statistics

Data were analyzed with SAS [11] using Spearman correlations, chi-square and logistic regression. Unless indicated otherwise, medians (10th–90th percentiles), or mean ± standard deviations, are indicated. Results are given as exact *p*-values as suggested by the American Statistical Association [12].

## 4. Results

### 4.1. Case Reports

Case 1. A 36-year-old woman was referred because of a suspicion of a ruptured ectopic pregnancy at 5 weeks of amenorrhoea. She had increasingly severe acute lower abdominal pain (7/10) for 24 h but no vaginal bleeding. Besides two uneventful pregnancies, her gynaecological, medical or surgical history was uneventful. The clinical exam was suggestive of abdominal bleeding because of abdominal tenderness and guarding, in a pale woman with tachycardia (109/min). She, however, was vitally stable with a BP of 101/63 mm Hg. A transvaginal scan confirmed an abdomen full of fluid suspected of being clotted blood up to Morison’s pouch. The haemoglobin was 8.3 g/dL. The urine pregnancy test was positive, but the uterus was empty. The coagulation profile was normal. Although the adnexae were not visualized, a ruptured ectopic pregnancy was suspected and an emergency laparoscopy was performed. The hemoperitoneum was confirmed but both oviducts were normal. The right ovary was bleeding from a small 1 × 1 cm lesion with an everted edge, which was not recognised as an ovarian pregnancy (Figure 2). This lesion was excised and two units of packed red blood cells and two units of fresh frozen plasma were given. The postoperative period was uneventful, and the day after surgery the β-HCG had dropped from 2923 international units (IU) to 1018 IU, becoming negative by day 12. The histopathological examination of the excised lesion found trophoblastic cells diagnosing the excised lesion as an ovarian pregnancy, which had not been recognised during surgery.

Case 2. A 33-year-old woman was referred because a ruptured ectopic pregnancy was suspected with an amenorrhoea of 32 days; she had acute abdominal pain, giddiness and shoulder tip pain for 8 h but no vaginal bleeding. Besides two vaginal deliveries, her antecedents were negative. She was pale, with severe pain (5/10), but the blood pressure and pulse rate were normal at 111/65 mmHg and 83/min, respectively. The abdomen was distended with tenderness and guarding. The transvaginal ultrasound scan found an empty uterus but a left adnexal mass of 3 × 3.5 cm and moderate fluid in the pelvis. Considering an HCG of 9693 mIU/mL, Hb of 8.7, PLT 257 and normal coagulation, an emergency laparoscopy was performed. Besides a 1000 mL hemoperitoneum, the fallopian tubes were normal and the left ovary was suspected of a ruptured ovarian pregnancy (Figure 2). The lesion was resected, hemostasis was performed and two units of blood transfusion were given. The postoperative recovery was uneventful and after 2 days, HCG had dropped to 4200 mIU/mL. The pathology report confirmed the ovarian pregnancy.

### 4.2. Review

#### 4.2.1. Study Selection, Study Characteristics and Risk of Bias

All publications on ovarian pregnancy from which data of the individual women could be retrieved were case reports, and thus without risk of bias.

#### 4.2.2. Synthesis of Results

Although not stated explicitly in all case reports, the diagnoses of ovarian pregnancy were all confirmed by pathology. The age of the women ranged from 16 to 45 years (median, 10th–90th percentiles being 30, 23–38), gravity from 1 to 9 (2.6, 1–5), parity from 0 to 7 (1, 0–3) (Appendix A) and abortions from 0 to 4 (0.4, 0–1). The frequency of previous ectopic pregnancies, IVF, IUI, tubal surgery, endometriosis, pelvic surgery, PID or IUD is shown in Figure 3. Surprisingly, six women had blocked or no fallopian tubes [13,14,15,16,17,18]. In addition, the incidences of IVF, IUD and PID seem rather high. A history of endometriosis was associated (Spearman) with a history of PID (*p* = 0.0078) or IUD (*p* = 0.0074), but not with ovarian pregnancies.

Most ectopic pregnancies were diagnosed within 8 weeks of amenorrhoea, but 15% of pregnancies had a duration of up to 44 weeks (Figure 4). Surprisingly, in 10 cases symptoms started before the end of a menstrual cycle (before 28 days of amenorrhoea). HCG concentrations were mentioned in only 60 cases and not surprisingly, a later diagnosis was associated with slightly higher concentrations of HCG (*p* < 0.0001). Symptoms were not specific with pelvic pain, vaginal bleeding and hypovolemic shock occurring in 77.4%, 45.2% and 11.5%, respectively (Figure 5). Ultrasound imaging was performed in 116 women, and a gestational sac was found in only 24 (21%) and a fetal pole in six. MRI was performed in four women, confirming an ovarian pregnancy in one only. Pregnancy was confirmed by HCG concentrations in 50%, and unknown in 43%, but negative in two cases (2%) [19,20].

Rare cases were one woman with a bilateral ovarian pregnancy [21], another with a left ovarian pregnancy diagnosed 3 weeks after left salpingectomy for tubal pregnancy after ovulation induction [22], two women with a tubal pregnancy and an ovarian pregnancy on the other side [23,24] and six ovarian pregnancies associated with an intra-uterine pregnancy (heterotopic pregnancy), two of them spontaneously [25,26], one after ovulation induction [27], one after IUI [28] and two with IVF [14,29]. In five of them, the intra-uterine pregnancies continued uneventfully after the excision of the ovarian pregnancy. One woman had an intra-uterine pregnancy and bilateral corpus luteum. After a miscarriage at 7 weeks, a laparoscopy for persisting lower abdominal pain revealed a ruptured ovarian pregnancy [25].

The site of the corpus luteum was mentioned in 22 reports only. However, five (22%) ovarian pregnancies occurred in the controlateral ovary [25,27,29,30,31], three of them being heterotopic [25,27,29].

Medical treatment was attempted in eight women (7.2%) without specific criteria of HCG concentrations, pelvic masses or fetal cardiac activity. Hyperosmolar glucose was given in one case in the lesion. Methotrexate was injected either locally or intramuscularly and was successful in five cases [22,27,32,33,34] but failed in two cases and surgical treatment was needed. In the first case [35], HCG was 3495 with a gestational sac of 7 × 6 × 7 mm and 3 days after a single dose of methotrexate, fetal heart activity was detected and HCG had increased to 5148. In the second case [36], the ectopic pregnancy mass size was 9 × 11 mm, the HCG concentration remained unchanged notwithstanding multiple doses of methotrexate, and the patient became unstable with severe abdominal pain.

The majority of women (95%) underwent surgery (Figure 6). Not surprisingly, more advanced pregnancies were treated more frequently with a laparotomy given the positive Spearman correlation of gestational age with surgery by laparotomy (*p* = 0.0130), and with less conservative surgery and more adnexectomies (*p* = 0.0001). Laparoscopic surgery was more frequent in women with a history of endometriosis (*p* = 0.0089) or IVF (*p* = 0.0005). Laparoscopies were performed in 43%, 50%, 47% and 10% for durations of amenorrhoea of <6 (n = 47), 7–8 (n = 32), 9–10 (n = 17), and more than 10 (n = 10) weeks of gestation, respectively. Adnexectomies instead of conservative surgery were performed in 30%, 6%, 41% and 78%, respectively. Not surprisingly, women in shock had more adnexectomies (*p* < 0.0001) In two women, a hysterectomy was performed (1.8%), one with a gestation of 44 weeks with the placenta involving the uterus [37], and one in a 45-year-old lady [38]. Follow-up after surgery was uneventful in all except one woman, to whom medical treatment was given because an HCG of 2134 IU/mL suggested residual trophoblastic tissue [39].

## 5. Discussion

### 5.1. Main Findings

Ovarian pregnancies seem to be associated with similar causal factors as extrauterine pregnancies, such as a history of PID or tubal surgery [1,6]. However, some aspects are unexpected, such as ovarian pregnancies in the contralateral ovary as the corpus luteum, similar to the observations of contralateral tubal pregnancies [40,41]. In addition, ovarian pregnancies in women without or with blocked oviducts are surprising, and micro fistula between the fallopian stump and peritoneal cavity to permit sperm transport seems rather speculative [15,16,17]. However, the diagnosis of blocked oviducts was rarely specified. Therefore, this may be an overestimation since the diagnosis by hysterosalpingography or methylene blue is often erroneous.

The diagnosis of ovarian pregnancy is not specific, and ovarian pregnancy should therefore be considered in all women suspected of extrauterine pregnancy with or without vaginal bleeding. The diagnostic accuracy of imaging such as ultrasound or MRI is insufficient to exclude an ovarian pregnancy and most are unexpectedly found during surgery. Most ovarian pregnancies are diagnosed in the first 8 weeks of pregnancy but surprisingly, some occur before a missed period and some are diagnosed as late as after 44 weeks of gestation. The suggestion that ovarian pregnancies are associated with more blood loss than tubal pregnancies could not be confirmed [2,3,13,18]. Occasionally, an ovarian pregnancy can result in choriocarcinoma [42]. In summary, the overall risk of an ovarian pregnancy ranges in pregnant women from 0.014% to 0.0025%, and in women with a suspicion of ectopic pregnancy because of symptoms or an adnexal mass, from 0.5% to 1%. The accuracy of diagnosis of ovarian pregnancy by imaging is insufficiently documented to judge false negatives.

Laparoscopy and laparoscopic surgery are the preferred techniques to diagnose and treat ectopic and ovarian pregnancies in women with intra-abdominal bleeding or a suspected ovarian mass, since they are minimally invasive, often permitting day surgery, and provide microsurgical techniques with minimal risk of ovarian damage and postoperative adhesions. If surgical experience is adequate, laparoscopy can even be performed in the most unstable patients. However laparoscopic surgery can be difficult if the bleeding is hidden in severe adnexal adhesions. In addition, the laparoscopic recognition of a bleeding ovarian lesion as an ovarian pregnancy and the differentiation from a bleeding corpus luteum [1] can be difficult, as illustrated in Figure 2. It is not clear when a uterine manipulator can be used, considering that a beginning intra-uterine pregnancy is difficult to exclude. Surgical excision of the ectopic pregnancy requires experience to limit ovarian damage by removing tissue or by coagulation. The results of medical treatment are variable. Care should be taken not to damage the corpus luteum during surgery, since a co-existing intra-uterine pregnancy was reported to continue uneventfully.

### 5.2. Comparison with Existing Literature

These data confirm and extend the literature which is limited to case reports and small series. The diagnosis is difficult and most ovarian pregnancies are unexpected findings during surgery [1].

Unexpected are the relatively frequent occurrences of ovarian pregnancies before a delay in menstruation, and in women with blocked tubes, and the association with intra-uterine pregnancies.

### 5.3. Strengths and Limitations

This is the largest series of ovarian pregnancies, permitting us to judge better incidences and associations. Unfortunately, these data do not improve diagnosis or judge ovarian-sparing surgery.

## 6. Conclusions and Implications

In conclusion, the diagnosis of ovarian pregnancy remains not specific and surgery is the treatment of choice. However, clinicians should be aware that ovarian pregnancies have to be suspected in all women with abdominal bleeding since ovarian pregnancies can occur before a delay in menstruation, even in women with blocked tubes. Since they occur in association with intra-uterine pregnancy, care has to be taken not to interrupt an eventual intra-uterine pregnancy with a uterine cannula and not to damage the corpus luteum during surgery.

## Figures and Tables

**Figure 1 jcm-12-01138-f001:**
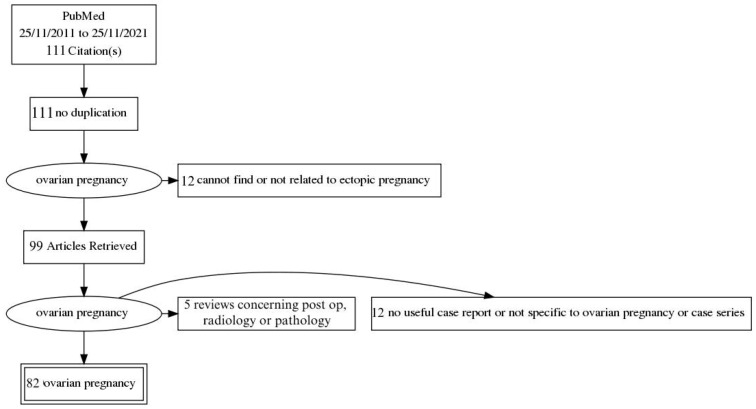
Prisma flow diagram for ovarian pregnancy search in PubMed from 2011 to 2021.

**Figure 2 jcm-12-01138-f002:**
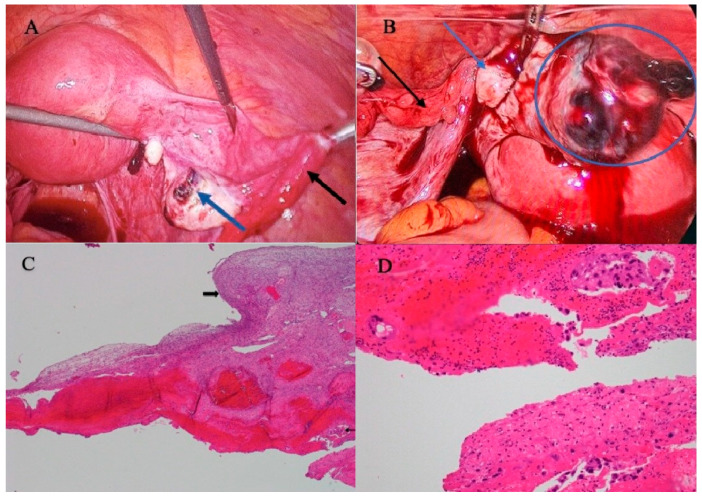
Case 1 illustrates that the laparoscopic recognition of an ovarian pregnancy can be difficult. (**A**) A normal right fallopian tube (black arrow), and on the right ovary, a lesion which was not recognised as an ovarian pregnancy (blue arrow). (**B**) An enlarged picture with a normal right oviduct (black arrow), and a right ovary (blue arrow) with a lesion not recognised as a ruptured ovarian pregnancy (blue circle). (**C**) On the histological examination of the excised lesion, a cyst-like wall lined by fibrinous hemorrhagic material is seen with scattered trophoblastic nests (thin arrow) overlying the ovarian edematous stroma with a primordial follicle (thick arrow) and covered with a mesothelial lining (hematoxylin and eosin (H&E) stain, original magnification ×40). (**D**) Foci of trophoblastic elements are embedded within a fibrinous inflamed decidualized tissue (H&E, ×200).

**Figure 3 jcm-12-01138-f003:**
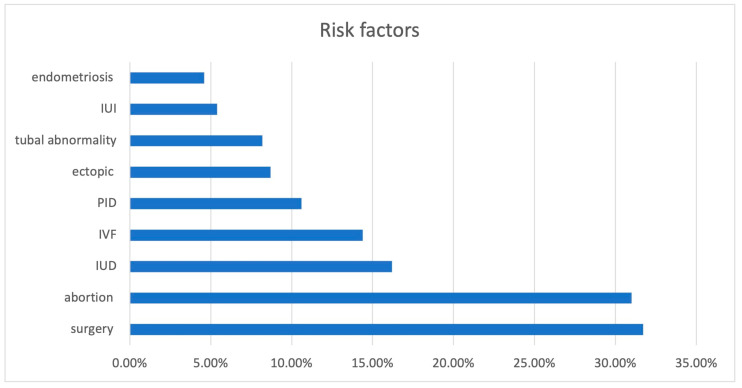
Incidences of previous ectopic pregnancies, IVF, IUI, tubal surgery, endometriosis, pelvic surgery, PID or IUD in women with ovarian pregnancies.

**Figure 4 jcm-12-01138-f004:**
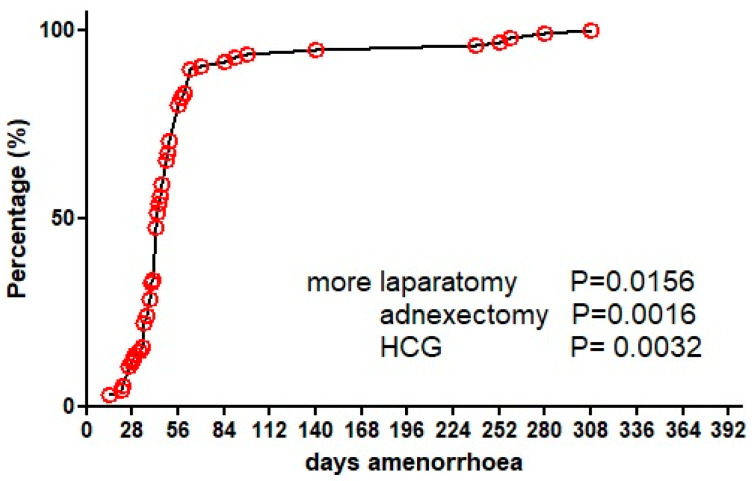
Duration of amenorrhoea in ovarian pregnancies.

**Figure 5 jcm-12-01138-f005:**
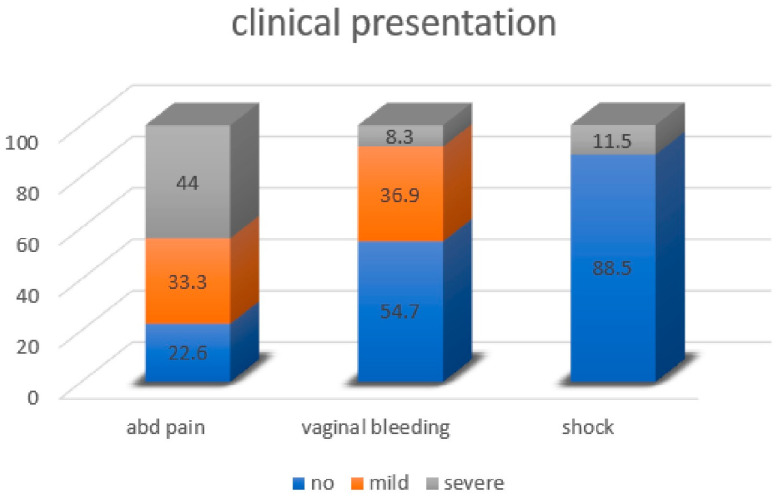
Presenting symptoms of ovarian pregnancies.

**Figure 6 jcm-12-01138-f006:**
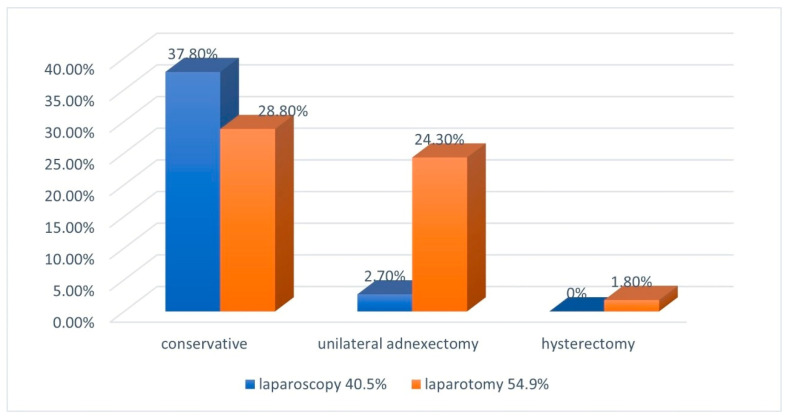
Treatment of ovarian pregnancies.

## Data Availability

Original data are available upon simple request to the corresponding author.

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
