# Peer review of "Ovarian Pregnancy: 2 Case Reports and a Systematic Review"

_jcm, 2023, doi:10.3390/jcm12031138_

Round 1

Reviewer 1 Report

Dear Authors,

Thank you very much for submitting your manuscript.

First of all abstact must be clearly divided into sections, for example line 11 Data Sources..

Furthermore, syntax and grammatical errors must be corrected such as line 14-15.

Syntax improvement in lines 19-22.

Line 32. Delete the space before 0,5%.

Line 36. Reference 9 must be referred in the end of the sentence.

Line 44. Better pathophysiology. Improve the expression.

Line 79. Improve syntax.

Line 82-83. Improve the expression.

Line 85-86. Improve expression.

line 93. To our surprise? Define.

Line 100. Bedside. Is it needed to the paper?

Line 169. More advanced pregnancies. After which week?

Conclusion. Please extend it with 3-4 lines, referring to the results of your survey.

Thank you very much in advance.

Yours sincerely.

Author Response

Thank you for your constructive comments.

Please find attached a highlighted and clean revision. All lines referred to will be those in the revised manuscript. 

First of all abstact must be clearly divided into sections, for example line 11 Data Sources..

Unfortunately, something must have gone wrong during editing. The uploaded pdf was structured as follows

 Objective 

Ovarian pregnancy is a rare but well-known pathology. However, pathophysiology, diagnosis and treatment are not 17 clearly established. 18

Data sources. All case reports published in Pubmed from Nov 2011 till Nov 2022. 19

Study eligibility criteria. A systematic review and 2 case reports of ovarian pregnancy 20

Study appraisal and synthesis methods. Not applicable 21

Results 22

Ovarian pregnancies occur in 8% of women without or blocked oviducts and in 23% on the other side than the corpus 23 luteum and the symptoms of ovarian pregnancies are not specific. Therefore ovarian pregnancy has to be suspected in 24 all women with abdominal bleeding. As for extrauterine pregnancies, the presence of an intrauterine pregnancy does 25 not rule out an ovarian pregnancy. 26

Surgical excision is the preferred treatment. Important is that in women with both an intra-uterine and an ovarian 27 pregnancy, care should be taken not to damage the corpus luteum. 28

Conclusions 29

Ovarian pregnancies can occur in women with blocked tubes, on the other side of the corpus luteum, in the presence of 30 an intrauterine pregnancy, and even when pregnancy tests or tranvaginal ultrasonography are negative.

The diagnosis 31 J. Clin. Med. 2023, 12, x FOR PEER REVIEW 2 of 13

being difficult to exclude, a laparoscopy is indicated in all women with intra-abdominal bleeding, keeping in mind that 32 an intra-uterine pregnancy cannot be exluded and that a corpus luteum need to be respected.

I have changed the manuscript back to the original 

Furthermore, syntax and grammatical errors must be corrected such as line 14-15.

 L15 Added: with

Syntax improvement in lines 19-22.

Conclusions were expanded as follows “Conclusions: Ovarian pregnancies can occur in women with blocked tubes. They occur on the other side of the corpus luteum. They can be associated with an intrauterine pregnancy, and the pregnancy tests or tranvaginal ultrasonography can be negative. Since the diagnosis is difficult to exclude, a laparoscopy is indicated in all women with intra-abdominal bleeding. However, the surgeon should realise that an associated intra-uterine pregnancy cannot be exluded and that therefore care should be taken not to damage the corpus luteum”

Line 32. Delete the space before 0,5%.

done

Line 36. Reference 9 must be reoneferred in the end of the sentence.

 ok

Line 44. Better pathophysiology. Improve the expression.

Revised as follows: The aim of the review was to understand better the pathophysiology of ovarian pregnancies and to improve diagnosis and therapy

Line 79. Improve syntax.

She had increasingly severe acute lower abdominal pain (7/10) for 24 hours but no vaginal bleeding

Line 82-83. Improve the expression.

The clinical exam was suggestive of abdominal bleeding since abdominal tenderness and guarding, in a pale woman with tachycardia (109/min). She, however, was vitally stable with a BP of 101/63 mm Hg

Line 85-86. Improve expression.

A transvaginal scan confirmed an abdomen full of fluid suspected of being clots up to Morison`s pouch. The haemoglobin was 8.3 g/dl. The urine pregnancy test was positive, but the uterus was empty

line 93. To our surprise? Define.

The histopathological examination of the excised lesion found to our surprise trophoblastic cells confirming an ovarian ectopic pregnancy, which had not been recognised during surgery

Line 100. Bedside. Is it needed to the paper?

 deleted

Line 169. More advanced pregnancies. After which week?

Not surprisingly more advanced pregnancies were treated more frequently with a laparotomy given the posiive Spearman correlation of gestational age with surgery by laparotomy ( P=0.0130.), and with  less conservative surgery and more adnexectomies ( P=0.0001).

Conclusion. Please extend it with 3-4 lines, referring to the results of your survey.

In conclusion, the diagnosis of ovarian pregnancy remains aspecific and surgery is the treatment of choice. However, clinicians should be aware that ovarian pregnancies has to be suspected in all women with an abdominal bleeding, since ovarian pregnacies can occur before a delay in menstruation, even in women with blocked tubes. Since they occur in association with an intra-uterine pregnancy care has to be taken not to interrupt an eventual intra-uterie pregnancy with an uterine cannula and not to damage the corpus luteum during surgery. 

Reviewer 2 Report

Thank you for giving me an opportunity to review the interesting manuscript entitled: Ovarian pregnancy: a systematic review and 2 case reports. While the cases are unique and may be worth publishing, the manuscript should be corrected before further consideration for publication. 

1. Abstract: correction in line with JCM recommendations is needed, it should be simplified and clarified, it is partially incoherent. 

2. Results: case presentation. If there are any ultrasound scans of presented cases, they should be added to better illustrate clinical situation. Figure 2 description should be corrected. Case presentation in the main text is not the same as figure description. Pictures illustrating each case should be separate (as case 1 and case 2). 

3. Synthesis of results: numbers in the brackets should be explained. What was the rate of histopathological confirmation of ovarian pregnancy according to the reviewed literature? What kind of medical treatment according to the literature was analyzed? How the agents were administered? 

4. Discussion: What were possible explanations of described phenomenon of ovarian pregnancy after excision or blockade of both tubes? What techniques were used for blocking tubes according to the reviewed literature? Are there any specific signs of ovarian pregnancy in imaging modalities? Maybe it is worth to propose diagnostic/therapeutic algorithm based on reviewed literature to clarify the management of ovarian pregnancies? Please explain and underline why minimally invasive surgical techniques are preferred and best option in such cases.

5. Language, syntax and spelling should be verified by the native English speaker.

Author Response

Thank you for your comments

  1. Abstract: correction in line with JCM recommendations is needed, it should be simplified and clarified, it is partially incoherent. 

Ze do apologise for having missed the instructions to authors for the abstract. The abstract therefore was extensively revised with the limit of 200 words.

In addition the text was checked again with grammarly for mistakes and double spaces. Since a Grammarly check was done on the original manuscript, something must have gone wrong when transferring to the JCM template.

  1. Results: case presentation. If there are any ultrasound scans of presented cases, they should be added to better illustrate clinical situation. Figure 2 description should be corrected. Case presentation in the main text is not the same as figure description. Pictures illustrating each case should be separate (as case 1 and case 2). 

We do thank for pointing this out and we do apologise for the mistakes. The case presentations were revised and the legend of Fig 2 was corrected.

  1. Synthesis of results: numbers in the brackets should be explained. What was the rate of histopathological confirmation of ovarian pregnancy according to the reviewed literature? What kind of medical treatment according to the literature was analyzed? How the agents were administered? 

Although indicated in statistical analysis it was repeated that in brackets medians, 10th - 90th percentiles are given.

Although not stated explicitly in all case reports, the diagnoses of ovarian pregnancy were all confirmed by pathology.  

Ultrasound imaging was performed in 116 women, and a gestational sac was found in only 24 (21%) and a fetal pole in 6. MRI was performed in 4 women confirming an ovarian pregnancy in 1 only

Medical treatment was attemted in 8 women (7.2%) without specific criteria of  HCG concentrations, pelvic masses or fetal cardiac activity. Hyperosmolar glucose was given in 1 case in the lesion. Methotrexate was injected either locally or intramuscularly and  was successful in 5 cases (22, 27, 32-34) but failed in 2 cases and surgical treatment was needed

  1. Discussion: What were possible explanations of described phenomenon of ovarian pregnancy after excision or blockade of both tubes? What techniques were used for blocking tubes according to the reviewed literature? Are there any specific signs of ovarian pregnancy in imaging modalities? Maybe it is worth to propose diagnostic/therapeutic algorithm based on reviewed literature to clarify the management of ovarian pregnancies? Please explain and underline why minimally invasive surgical techniques are preferred and best option in such cases.

We added; However, the diagnosis of blocked oviducts was rarely specified. Therefore this may be an overestimation since the diagnosis by hysterosalpingography or methylene blue is often erroneously wrong

In summary the overall risk of an ovarian pregnancy ranges in pregnant women from 0.014% to 0.0025%, in women with a suspicion of an ectopic pegnancy because of symptoms or an adnexal mass from 0.5% to 1%. Accuracy of diagnosis of ovarian pregnancy by imaging is insufficiently documented to judge false negatives.

Laparoscopy and laparoscopic surgery are the preferred techniques to diagnose and treat ectopic and ovarian pregnancies in women with intra-abdominal bleeding or a suspected ovarian mass, since minimally invasive permitting often day surgery and provided microsurgical techniques with minimal risk of ovarian damage and postoperative adhesions. If surgical experience is adequate laparoscopy can even be performed in most unstable patients. However laparoscopic surgery can be difficult if the bleeding is hidden in severe adnexal adhesions. Also, the laparoscopic recognition of a bleeding ovarian lesion as an ovarian pregnancy and the differentiation from a bleeding corpus luteum(1) can be difficult as illustrated in Figure 2.   

  1. Language, syntax and spelling should be verified by the native English speaker.

I (PK) reviewed the text personally and runned a Grammarly check at the end.

Round 2

Reviewer 2 Report

After implementation of reviewers suggestions the manuscript has sufficient educational and scientific value. Therefore I recommend it for further steps of publication process.